# Variation in Hydric Response of Two Industrial Hemp Varieties (*Cannabis sativa*) to Induced Water Stress

Hang Duong [1] , Brian Pearson [1], Steven Anderson [1], Erin Berthold [2] and Roger Kjelgren [1,*]

[1] Mid-Florida Research and Education Center, University of Florida, Apopka, FL 32703, USA
[2] College of Pharmacy, University of Florida, Gainesville, FL 32611, USA
* Correspondence: rkjelgren@ufl.edu

**Abstract:** Information on industrial hemp (*Cannabis sativa*) water use and water stress is sparse. We studied water stress impact in two essential-oil hemp cultivars ('Wife' and 'Cherry') prompted by anecdotal differences in growth and water use. In a greenhouse setting, we measured water relations, water use, growth, and essential oil (CBD-cannabidiol and THC-delta-9 tetrahydrocannabinol) concentrations. Water stress did not significantly affect THC and CBD concentrations, but both cultivars responded to water stress by reducing transpiration through notably different mechanisms. 'Cherry' had more anisohydric behavior, maintaining high stomatal conductance (Gs) and more negative leaf water potential until root zone water depletion triggered partial afternoon stomatal closure to moderate stress, resulting in lower flower and CBD yield. By contrast, water-stressed 'Wife' rapidly defoliated half its leaf area in balance with less applied water and so maintained high Gs and flower yield on par with well-watered plants, suggesting potential for deficit irrigation to conserve water and reduce post-harvest vegetation management. Differences in water use translated to provisionally suggested crop coefficients of 1 for 'Cherry' and 1.3–1.5 for 'Wife', but further research is needed. Because hemp is genetically diverse, and cultivar naming conventions are currently lax, further germplasm screening and research are needed to determine the extent to which either conservative 'Cherry' or the water-stress defoliation response of 'Wife' is found in the larger population of hemp cultivars.

**Keywords:** *Cannabis sativa*; water use; industrial hemp; cannabidiol





## 1. Introduction

In 2018, the United States Congress passed the Agriculture Improvement Act of 2018, federally legalizing industrial hemp (*Cannabis sativa* L. < 0.3% delta-9 tetrahydrocannabinol-THC) production, which triggered intense interest by growers seeking an ostensibly lucrative market. Industrial hemp is a genetically diverse, annual, dioecious, short-day (flowering) subshrub [1–4], a unique horticultural combination. Hemp is also a multi-use crop grown for fiber, seed, and essential oils/cannabinoids, which have potential medicinal applications [5]. An ancient crop, hemp has been cultivated for thousands of years and is potentially a significant opportunity for modern agriculture [6].

Hemp's future as an economically viable and environmentally sustainable crop, particularly cannabinoid-type hemp, depends on research into operational practices. Historically, research has been limited due to legal restrictions [7]. Largely unknown are optimum inputs of nutrients and water for the most growth and yield with the fewest environmental consequences. A number of studies have reported responses to N rates among [8], but water-related studies have been narrowly focused on cannabinoid concentration [9,10] or screening for yield variability among cultivars [1,11], but less so to inform operational practices. Four practice-related questions linger regarding physiological and production responses to controlled water stress in essential oil/cannabinoid hemp: (1) To what degree does hemp regulate water relations via transpiration/stomatal opening/closing to moderate internal water potential; (2) What is hemp's water use rate relative to reference evapotranspiration (ET) under variable water inputs given varied stomatal control;

(3) What is the effect of stress on flower yield and concentrations of key cannabinoid compounds of interest, particularly THC and CBD; and (4) What is the genetic variation in yield, concentration, and water use/relations amongst hemp genetic material?

Peer-reviewed information on water relations of hemp is sparse. High water-use rates imply high transpiration driven by open stomates [12]. However, the extent internal water potential and photosynthesis are mediated by a stomatal aperture in response to dry air and dry soil along the isohydric/anisohydric continuum [13] has not been reported for hemp. Other research has shown for fiber hemp stomatal conductance upwards of 400 mmol $m^{-2}s^{-1}$ with distinct midday depression in response to dry air, a range in conductance values similar to other crops [14,15]. Hemp is also reported to be sensitive to root zone water stress by reducing water demand by closing stomates short term and defoliating to reduce transpiration under longer water stress [16].

Information on hemp water use is critical for scheduling irrigation timing and amount to maximize yield with the least water. Studies of applied cannabis production suggest high irrigation requirements [17], but actual water use relative to ETo (reference evapotranspiration) is unknown for commercial cannabis production [12]. Extant studies of irrigated, field-grown essential oil/cannabinoid hemp within a limited range of ETo suggest a crop coefficient-Kc (in order to estimate crop water use as a fraction of ETo) value of 100% of ETo [11,13], while for field-grown fiber, researchers used variable growth-stage Kc values including 1.1 to 1.8 for two hemp cultivars [14]. Studies have not yet linked water relations/stomatal control to measured water use.

Hemp biomass and essential oil/cannabinoid production appear sensitive to water stress. Irrigating industrial hemp at rates 75–80% of ETo reduces biomass yield, varying with planting date and density [13], but well-watered and water-stressed biomass yield also varies widely across genotypes [11], particularly flower yield [1,2]. Water stress impact on concentration of key cannabinoids of interest is unclear but critical to growers because of the risk of THC levels exceeding the 0.3% federal threshold. The effect of water stress on essential oil/cannabinoid content is ambiguous. Anecdotal gray literature and one controlled study on high THC cannabis production [9] suggest that water stress elevated cannabinoid concentrations, but not for hemp in another greenhouse study [10] nor in field-grown hemp [13].

The genetic amplitude of hemp is vast [2] for key traits such as flowering photoperiod [18], CBD versus THC production [4,19], and nitrogen response [8]. The genetic amplitude of key cannabis traits in response to water appears messy: response to water stress in controlled environment settings with young, flowering cannabis [9], was found to have increased THC and CBD concentrations but other findings reported a decrease in both compounds, each study investigating a single cultivar [10]. In a field production setting, it was found that there were no consistent differences in THC and CBD concentrations between two well-watered and water-stressed essential oil/cannabinoid hemp cultivars [13]. Overall, yield in response to water stress also appears to vary widely among industrial hemp cultivars [1,2,11].

During other hemp research at our location, we anecdotally observed that the cultivar 'Cherry' grew slower and used less water than the cultivar 'Wife'. These observations prompted the present study as a snapshot of quantitative differences in water relations, growth and flower yield, water use, and cannabinoid concentration responses to water stress of these two distinct hemp cultivars.

## 2. Materials and Methods

### 2.1. Experimental Setup

This research was conducted at the University of Florida's Mid-Florida Research and Education Center in Apopka, Florida, from late February to March 2020. The study was located in a greenhouse under poly plastic that reduced incoming solar radiation by 30% and temperature moderated by a large evaporative cooler. Solar radiation, relative humidity, and air temperature were monitored in the greenhouse using a datalogger

with photosynthetically active radiation (PAR), dew point, and air temperature sensors (WatchDog 2475; Spectrum Technologies, Inc., Aurora, IL, USA) mounted on a post 2 m above the floor inside the greenhouse. Vapor pressure deficit (VPD) was calculated from air temperature and dew point data.

Two hemp cultivars, 'Wife' and 'Cherry', were studied, each certified by the University of Florida industrial hemp cultivar approval program (ANOWIFE001 and ANOCHERRY001, respectively) obtained from ANO Hemp LLC, Parachute, Colorado, United States. Beyond this information, nothing further was known of their parentage apart from 'Cherry' being a common name in hemp cultivars [4] while 'Wife' is less common.. We selected these two cultivars because they observationally differed in morphology and in anecdotal water use, so they possibly offer differences in water stress response. 'Wife' was taller and faster growing with slender *C. sativa*-type leaves and required frequent irrigation, while 'Cherry' was shorter and slower growing with wider C. indica-type leaves [2] and required less frequent irrigation.

Eight cuttings were taken from mother stock plants of each cultivar in early November 2019 and rooted using a hydroponic misting system. Once rooted in mid-December, plants were transplanted to 5.7-L containers filled with a commercial substrate (Promix HP-70% peat moss: 30% perlite with mycorrhizae, Premier Tech Horticulture, QC, Canada), fertilized with 50 g of 6-month slow release Osmocote 15-9-12 (Scotts, Marysville, OH, USA) and allowed to establish in the study greenhouse and irrigated to saturation with a spray stake at predawn each morning. To prevent flowering, plants were kept under 1000 W metal halide lamps at 110,000 lumens for 22 h light and two hours dark, in addition to seasonal ambient light with a natural sunlight duration of approximately 10 to 11 h a day. Plants were topped in mid-January 2020 to bring all plants to a uniform height. On 30 January, plants were transplanted to 11-L containers in the same media, fertilized again at the same rate and type, and supplemental lighting was turned off to allow natural light to initiate eight weeks of flowering. One container plant of each cultivar × water treatment combination was randomly assigned to square blocks on a greenhouse table spaced 0.5 m apart, with four replicate blocks spaced a meter apart along the greenhouse table. This spacing allowed adequate ventilation to avoid boundary-layer interference among plants and blocks.

### 2.2. Data Collection

In mid-February, during early flowering stage, but prior to applying water stress treatments, a dawn-to-dusk measurement of stomatal conductance and water potential was conducted to characterize the daylight pattern of water relations under well-watered conditions. Stomatal conductance (Gs) was measured with a porometer (model AP4, Delta-T Devices, Cambridge, England) every two hours on three sunlit leaves per plant for the eight replicate plants of each cultivar until early evening. Leaf water potential (LWP) was concurrently measured on one excised (measured immediately), fully exposed mature leaf per plant at mid-crown level starting at predawn and then continuing every two hours until early evening with a Scholander-type pressure chamber (model 3000, Soil Moisture Inc, Santa Barbara, CA, USA), where plants were measured immediately after excision.

On 28 February 2020, four weeks after flowering initiation, water stress treatments were applied to progressively dry the study plants. To track changes in substrate root zone volumetric water content, sensors were installed at a 45-degree angle downward across the plant root zone, at a density of one per pot/plant, and connected to a datalogger (model EC-5 probe and Zentra logger, Decagon Inc., Pullman, WA, USA) which was calibrated to organic media that recorded volumetric water content hourly. To quantify water use gravimetricaally we weighed each plant at 7 a.m. each morning at the predawn stage of the dry down. At that time, plants in the well-watered treatment were rewatered at 110% of the previous day's calculated water use (wet treatment).

Half the plants of each cultivar were progressively dried down over an eight-to-10-day period by changing irrigation levels to 80% of the previous day's water use (dry treatment),

similar to the procedure described by Tang [16]. Beginning the first day of the dry down in early March, Gs was measured daily during the estimated daily maximum level from 10:00 to 11:00 a.m., then again from 2:00 to 3:00 p.m. at the estimated point of maximum air temperature when stomatal closure was likely to occur. Concurrently, LWP was measured predawn and mid-afternoon together with Gs measurements following the previously described procedure. The dry down and water relations measurements were suspended at a steady-state level of water stress based on the dawn-to-dusk pattern of Gs and LWP previously established when daily Gs of water stressed plants fell to 50% of maximum values at the start of the dry down. All plants continued to be irrigated to replace the previous day's volume of water loss, thus maintained at relatively steady state water stress for the remainder of the study. Water-stressed (dry) and well-watered (wet) irrigation was maintained until eight weeks after flower initiation. At that point, the study was stopped, and all plants harvested. All leaves for each plant were separated from the other biomass, run through a leaf area meter (LI-3000; LI-COR, Inc. Lincoln, NE, United States), dried at 60 °C for 48 h, then weighed. Stalks and flowers were then dried for at least 48 h at 60 °C, and then the stalks were separated from the flowers, and each weighed.

### 2.3. Cannabinoid Analysis

The procedure followed for hemp dry matter sampling and analysis of low THC hemp for cannabinoid concentration was that of the Association of Official Analytical Chemists Standard Method Performance Requirements (AOAC SMPR) 2019.003, as described by Berthold et al. [20]. Six weeks and eight weeks after the vegetative to floral transition was initiated, 15 cm long apical floral samples (cola) were collected from each plant. Cola samples were dried using a forced cool air technique described by previous research for seven days [21]. Dried samples were ground into a fine powder using a coffee grinder and stored in 100 mL glass vials in preparation for analysis of cannabinoid content.

Extraction and quantification of cannabinoids were conducted on the University of Florida campus by the College of Pharmacy. Ground samples were weighed, and cannabinoids were extracted by adding a solution of methanol and water (95:5, $v/v$) acidified with 0.005% formic acid at a 1:100 $w/v$ plant material to solvent concentration ratio. The solution was vortex mixed for 5 min, sonicated for 5 min, and centrifuged at 4 °C, 3220× $g$ for 10 min. The supernatant was serially diluted using extraction solvent until the sample concentration fell within the quantification range. Quantification of cannabinoids was conducted using a Waters I-Class Acquity UPLC (Milford, MA, USA) coupled with a Waters Xevo TQ-S Micro™ triple-quadrupole mass spectrometer (MS/MS). Unfortunately, we were unable to obtain THC in the eight-week sample due to a drying malfunction. A full description of the analytical process can be found in Berthold et al. [20].

### 2.4. Statistical Analysis

Initial dawn-to-dusk water relations (GS, LWP) were plotted against hour (SigmaPlot 13, Systat Inc., San Jose, CA, USA) with standard deviation pooled across treatments, along with hourly PAR, daily maximum air temperature, and VPD. Differences between cultivars were compared with t-tests using (JMP®, Version 14. SAS Institute Inc., Cary, NC, USA). Dry down midmorning and midafternoon GS and predawn and afternoon LWP were then plotted against the day of dry down, and again with standard deviation pooled across treatments, and along with PAR and maximum daily air temperature. Differences among time of data collection and wet and dry treatments within a cultivar were compared with SAS-JMP using a two-way ANOVA and tested for equal variances. Post-hoc analysis, when significance was detected among treatments, was conducted with Tukey's HSD (honest significance difference) at a 5% level of significance. Predawn (at 7 a.m. prior to irrigation) volumetric substrate water content recorded with the sensor-datalogger system was extracted and averaged for each treatment, and standard deviation was pooled across wet/dry treatments for each cultivar. Dry volumetric water content was divided by wet water content, and the resulting ratio that tracked substrate water depletion was plotted

against a day of dry down along with pooled standard deviation. Differences between cultivars were again compared with *t*-tests using SAS-JMP. Finally, gravimetric water use in grams per day was converted to volume units with media bulk density, then to depth units by dividing by final leaf area. Standard deviation was pooled across treatments within a cultivar.

Differences in cannabinoid concentrations, biomass, ratio of flower to total biomass (yield index) and total cannabinoid (THC and CBD concentration × flower biomass) yield were analyzed again with SAS-JMP using a two-way ANOVA with two levels of cultivar and two levels of water stress treatments, test for equal variances. Post-hoc analysis, when significance was detected among treatments, was conducted as described with Tukey's HSD (honest significance difference) at a 5% level of significance.

## 3. Results

The hemp varieties 'Cherry' and 'Wife' responded very differently to water stress in terms of growth and water relations but less so for cannabinoid concentrations. During the month of March, when the dry-down study was conducted, day length increased rapidly, greenhouse maximum air temperatures ranged from 28 to 32 °C, and VPD levels were approximately 3 kPa.

### 3.1. Preliminary Water Relations

Daylight pattern for Gs and LWP of well-watered hemp was assessed to identify maximum and minimum values for later sampling during the dry-down study (Figure 1). Stomata opened rapidly in the morning with increased sunlight, as Gs for both varieties rose after dawn (around 8:00 a.m.) to near the daily maximum that was maintained through midday in both cultivars at approximately 500 mm m$^2$-s$^{-1}$.

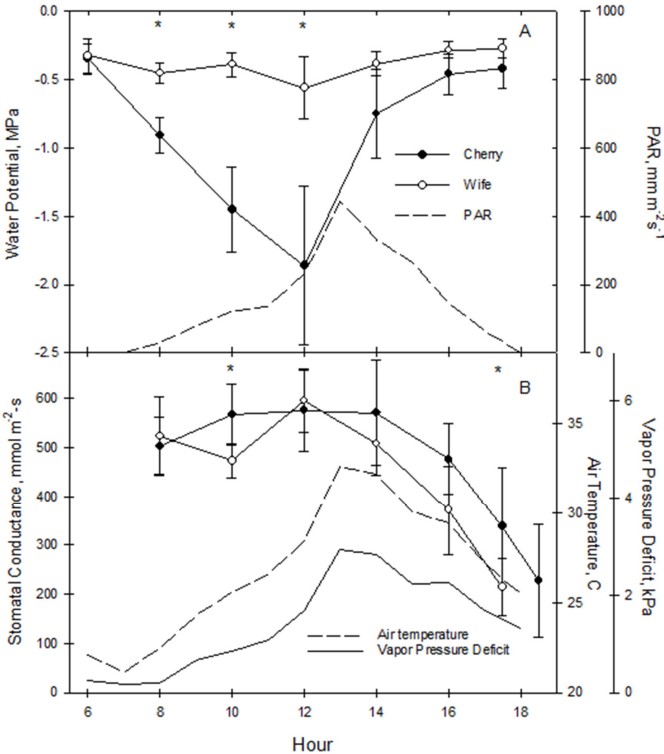

**Figure 1.** Dawn-to-dusk stomatal conductance (**A**) and leaf water potential (**B**) for two industrial hemp cultivars, 'Cherry' and 'Wife', on 19 February 2020, plus standard deviation (*n* = 8), Also shown are atmospheric variables photosynthetically active radiation (PAR), air temperature, and vapor pressure deficit. Hourly data points with asterisks indicate significant differences between cultivars at *p* < 0.05.

The daytime LWP pattern differed greatly between the two cultivars. 'Cherry' LWP became more negative with increasing Gs due to internal stem resistance to water flow but rapidly became less negative from 2:00 p.m. onward as the Leaf water potential pattern of 'Wife' unexpectedly differed as it hovered between −0.3 to −0.5 MPa all day despite Gs essentially the same as 'Cherry'. This anomalous behavior suggested internal resistance to water flow was remarkably minimal. Based on these results, subsequent Gs readings were collected mid-morning and midafternoon between 1:00 to 2:00 p.m. to capture potential daily maximum and VPD-induced minimum, respectively, and LWP was measured predawn and early afternoon between 12:00 and 2:00 p.m for targeted maximum and minimum values.

### 3.2. Dry down Water Relations

During dry-down days one to six, morning and afternoon Gs patterns for both cultivars were similar in range to values observed in the preliminary study, largely between 400–600 mm $m^2$-$s^{-1}$ (Figure 2C,D). Stomatal conductance during this initial dry-down period did not differ between morning (a.m.) and afternoon (p.m.) Gs readings for either cultivar. Differences in LWP between cultivars and AM/PM readings were much greater, consistent with the initial study (Figure 2). Wet-treatment 'Cherry' LWP followed an expected pattern of maximum daily predawn at −0.2 to −0.3 MPa, then declined to the p.m. minimum of −1.0 to −1.4 MPa, likely as a result of internal vascular resistance to water flow in response to daily maximum evaporative demand (Figure 2A). Water potentials of dry-treatment 'Cherry' were more negative than wet 'Cherry' at predawn by several MPa during the initial eight days of drying, suggesting that dry-treatment 'Cherry' reacted quickly to decreased water availability but maintained high Gs (Figure 2C); water stress emerged on day 10 when only PM Gs in dry 'Cherry' fell to 50% of well-watered levels.

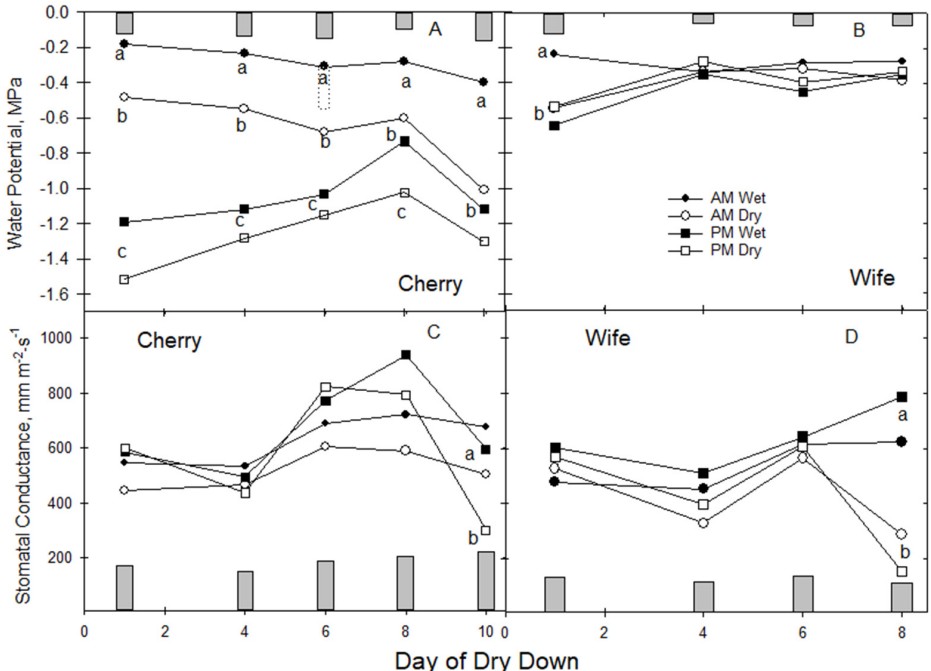

**Figure 2.** Preliminary dawn-to-dusk water relations study showing: graphs (**A,B**) predawn (AM) and midafternoon (PM) leaf water potential for the hemp cultivars 'Cherry' and 'Wife', respectively; graphs (**C,D**) predawn (AM) and mid-afternoon (PM) stomatal conductance for water-stressed and well-watered hemp varieties 'Cherry' and 'Wife'. Bars along the *X*-axis of each graph are pooled standard deviation across treatment and cultivar for each measurement day (*n* = 16). Data points within a day separated by different letters are significant at *p* < 0.05 using Tukey's posthoc HSD test (*n* = 4).

Not so for dry-treatment 'Wife' (Figure 2B,D). Water stress emerged abruptly on day 8 of the dry down, where both a.m. and p.m. Gs fell to 50% of the well-watered levels, unlike 'Cherry'. Concurrent a.m. and p.m. stomatal closure in 'Wife' suggested that it was very sensitive to root zone water stress, with insufficient nighttime capillary water movement to allow early morning stomatal recovery. It is possible that 'Wife' LWP was more negative, indicating water stress, on day seven and triggered stomatal closure on day eight that moderated a.m. and p.m. LWP. However, all four treatment combinations for 'Wife' LWP, a.m. versus p.m. and wet versus dry, showed no differences, staying constant between 2–6 MPa during the 8-day dry down. This response was quite different from 'Cherry' but echoed the pattern observed in the preliminary study that resistance to water flow in 'Wife' was negligible.

### 3.3. Root Zone Water Content

Our strategy of slowly decreasing water application to the dry plants was effective, as the initial dry-down phase days 1–8 saw a steady decline in relative root zone water content (ratio of water stress as a fraction of well-watered water content) that did not differ between the two cultivars (Figure 3B). Stomatal conductance data (Figure 2) indicated water stress onset in 'Wife' emerged on day eight at a root zone water content ratio of approximately 50% of the wet hemp plants (actual volumetric water content was 31% dry, 59% wet). On day nine, the relative root zone water content of 'Wife' dropped to less than 40% of the wet-treatment plants but not so for 'Cherry'. On day ten, the relative water content in dry-treatment 'Cherry' dropped to 35% from that of the wet plants. From day ten onward, enough water was added daily to maintain relative water-stressed root zone water content in 'Wife' at 30–40% of wet hemp and 'Cherry' at 40–50%). During the last ten days of the study, dry-treatment 'Cherry' relative root zone water content was consistently greater than that for 'Wife', suggesting Gs of 'Cherry' water-stressed plants may have declined to a point where further depletion of root zone water was avoided.

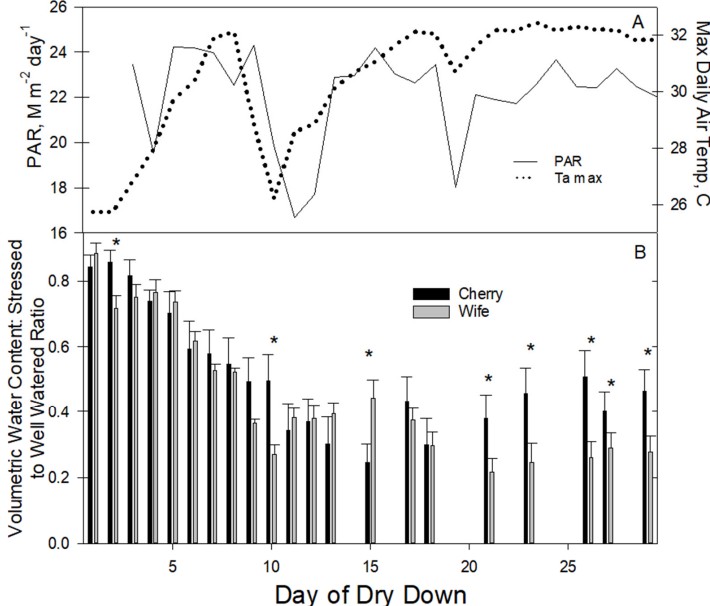

**Figure 3.** Dry down of two hemp varieties showing climate ((**A**); cumulative incoming daily photosynthetically active radiation [PAR] and maximum daily air temperature [Ta max]) and concurrent decline in root zone water content (**B**) of two varieties of container-grown hemp., Bars show the ratio water stressed water content as a fraction of well-watered plants for the two hemp varieties, 'Cherry' and 'Wife' during the study of progressive root zone water depletion and subsequent steady-state water stress. Standard deviation bars are pooled across wet and dry treatments within a cultivar ($n = 8$). Asterisks above the daily data point are significantly different at $p < 0.05$ ($n = 4$).

### 3.4. Growth and Flower Yield

Growth varied more between varieties than between well-watered and water-stressed plants (Table 1). Total and individual leaf area growth is an integrated measure of water stress response during vegetative growth, as less of both reduces transpiring leaf area and total plant transpiration that, slows the depletion of root zone water. The leaf areas of both wet- and dry-treatment 'Cherry' were not different, which was not unexpected given that shoot elongation and non-florescent leaf growth stopped with the onset of flowering and water stress treatments, and both were also not different from wet 'Wife'. However, the leaf area of dry-treatment 'Wife' was less than half that of the wet treatment due to drought-induced defoliation. Dry 'Wife' senesced and shed its leaves rapidly beginning on day 8 in response to water stress, a mechanism to reduce transpiring leaf area and root zone water depletion.

**Table 1.** Morphological and biomass responses to induced water stress for two varieties of industrial hemp, Wife and Cherry, including leaf density expressed as total leaf area per plant, total flower yield, and yield index defined as the ratio of flower weight to total biomass.

| Variety [1] | Treatment | Leaf Area (cm$^2$) | Specific Leaf Area (cm$^2$g$^{-1}$) | Flower Yield (g) | Yield Index |
|---|---|---|---|---|---|
| 'Cherry' | Wet | 3881 ± 1415 a | 180 ± 22 a | 64 ± 7 a | 0.50 ± 0.04 a |
| 'Cherry' | Dry | 2929 ± 533 a | 193 ± 8 a | 43 ± 7 b | 0.50 ± 0.06 a |
| 'Wife' | Wet | 2734 ± 237 a | 153 ± 14 b | 75 ± 6 a | 0.51 ± 0.01 a |
| 'Wife' | Dry | 1257 ± 306 b | 118 ± 15 c | 66 ± 9 a | 0.54 ± 0.02 a |

[1] Total leaf area per plant, specific leaf area, flower yield, and yield index (dry flower weight divided by total dry weight flowering for wet and dry treatments of the two hemp cultivars 'Cherry' and 'Wife' (*n* = 4). Values within a column separated by different letters are significant at *p* < 0.05 using Tukey's post-hoc HSD test.

Interestingly, 'Cherry' leaves were less dense than those of 'Wife'. The specific leaf area (SLA) of 'Cherry' did not differ between wet and dry plants, with wet 'Cherry' possessing approximately 7% greater individual leaf area). However, 'Cherry' SLA was 20% greater than wet-treatment 'Wife' and 40% greater than dry-treatment 'Wife', which translated to 'Cherry' investing in more area per unit carbon than 'Wife'.

### 3.5. Water Use

Information on how hemp utilizes water is vital in managing irrigation for maximum flower yield for the least amount of water (Figure 4). Over the study period that was mostly sunny with greenhouse temperatures from 28 to 32 °C (Figure 4A,B), VPD levels were consistently around 3 kPa (see Figure 1B), transpiring leaf area and Gs (regulating transpiration) governed water use for all treatment combinations, but was different between the two cultivars. Water use by weight of wet plants for both varieties increased over time, with 'Wife' about 25% greater than 'Cherry' over the latter half of the study (Figure 4C,D). Wet-treatment 'Wife' water use was approximately 2 L d$^{-1}$, while for wet 'Cherry', it was lower, approximately 1.5–1.6 L d$^{-1}$. For dry-treatment plants of both cultivars, water use over the latter part of the study ranged from 40% to 60% of the wet plants because of drought stress reducing transpiration for different reasons. The normalized leaf area (water weight divided by leaf area) gave very different responses (Figure 4E,F). Wet-treatment 'Cherry' water use over the last 15 days of the study was approximately 4 mm d$^{-1}$, while dry-treatment plants were 2.2 mm d$^{-1}$, likely due to stomatal closure in response to water stress (Figure 4E), similar to findings [16] in fiber hemp. Transpiration in 'Wife' normalized for leaf area was substantially higher than 'Cherry', from 6–8 mm d$^{-1}$ over the latter half of the study.

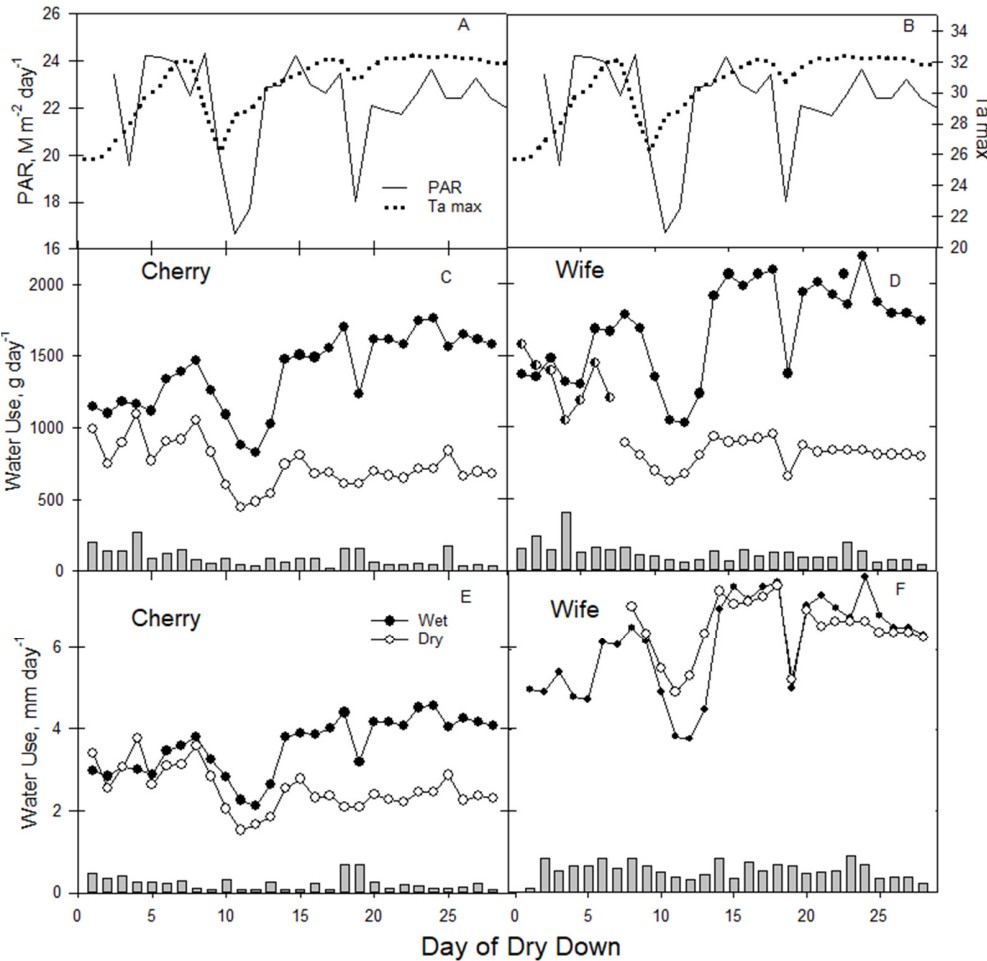

**Figure 4.** Water use of well-watered and water-stress hemp varieties 'Cherry' and 'Wife'. (**A**,**B**): daily cumulative PAR and daily maximum air temperature for each day of the dry down; (**C**,**D**): average daily water use in grams per plant for well-watered and water-stressed 'Cherry' and 'Wife' varieties; (**E**,**F**): water use in mm per day (weight normalized to transpiring leaf area) for well-watered and water-stressed 'Cherry' and 'Wife' varieties. In graph (**D**) half, circle data points represent days for dry 'Wife' plants prior to defoliation. Bars along the *X*-axis of each graph are pooled standard deviation across wet and dry treatments within a cultivar (*n* = 4 for each data point).

Although water use data was collected in a greenhouse where environmental conditions were dissimilar from ambient outdoor conditions that were late-winter, evaporative demand in the greenhouse based on air temperature and humidy would be approximately equivalent to environmental conditions experienced outdoors during early summer in Florida, where reference evapotranspiration (ETo) is approximately 4–5 mm d$^{-1}$. This approximation of early summer evaporative demand conditions suggested that the ratio of hemp water use to early summer reference evapotranspiration can provisionally approximate crop coefficients used determining when and how much to irrigate.

### 3.6. Cannabinoid Concentration and Yield

Water stress had a minimal impact on cannabinoid yield (Table 2). At six weeks of flowering, all treatments had THC concentrations above the federal limit of 0.3%, with 'Wife' slightly greater than 'Cherry', above the legal limit to harvest and sell as industrial hemp. THC concentrations did not differ meaningfully between wet and dry treatments at four weeks, Cannabidiol concentration did not vary consistently among treatments, as CBD did not differ among treatments at six weeks but was marginally higher in 'Cherry' at eight weeks into flowering at 10–11% concentration. Water stress impact on flower yield

resulted in large differences in total CBD yield. Wet-treatment 'Wife' plants produced the most CBD on both sampling dates (assuming the same flower yield for both dates), and dry-treatment 'Cherry' had the lowest yield due to low flower yield.

**Table 2.** Cannabinoid (THC—tetrahydrocannabinol—and CBD—cannabidiol) concentration and yield (flower weight x concentration) at four and eight weeks into flowering. Values within a column separated by letters are significantly different at $p < 0.05$ using Tukey's post-hoc HSD test ($n = 4$), plus standard deviation.

| | | Four Weeks | | Eight Weeks | Four Weeks | Eight Weeks |
|---|---|---|---|---|---|---|
| Variety [1] | Treatment | THC, % | CBD, % | CBD, % | CBD, g | CBD, g |
| 'Cherry' | Wet | 0.38 ± 0.07 b | 9.7 ± 1.4 a | 11.4 ± 0.07 a | 6.2 ± b | 7.3 ± 0.2 ab |
| 'Cherry' | Dry | 0.39 ± 0.01 b | 9.6 ± 0.2 a | 12.1 ± 0.5 a | 4.2 ± c | 5.3 ± 0.6 b |
| 'Wife' | Wet | 0.49 ± 0.05 a | 11 ± 0.07 a | 10.6 ± 1.4 ab | 8.4 ± a | 8.0 ± 1.2 a |
| 'Wife' | Dry | 0.40 ± 0.05 ab | 9.5 ± 0.8 a | 10.2 ± 0.4 b | 6.3 ± b | 6.8 ± 0.5 ab |

[1] Values within a column separated by different letters are significant at $p < 0.05$ using Tukey's posthoc HSD test.

## 4. Discussion

The initial dawn-to-dusk study (Figure 1) found that stomatal conductance reported here was somewhat higher than the range up to 300 mmol m$^{-2}$s$^{-1}$ reported by other researchers [14] for field-grown fiber hemp, but may be an artifact of hemp's genetic diversity [22]. Midday decline in Gs values for both cultivars is likely a combination of stomatal sensitivity to VPD seen in most woody and many herbaceous species [15] as VPD levels exceeded 2 kPa. Additionally, lower light may have contributed lower afternoon Gs values as photosynthetically active radiation fell to 200 mm m$^{-2-s}$ due to partial cloudiness during the afternoon. Midday minimum LWP of around −1.5 MPa was more negative than the 1.0 MPareported for cannabis elsewhere [9]. Afternoon stomatal closure likely moderated internal water tension, a typical pattern in most species, including subshrubs on the border between herbaceous and woody plants [23].

Imposed root zone water depletion showed "Wife" responded more quickly to water stress (Figure 2) than "Cherry, both morphologically and physiologically (Figure 3). For many plant species responding to progressive water stress (Figure 2), afternoon Gs initially declines under daily peak evaporative demand [15], moderating transpiration, internal water potential, and root zone water depletion (Figure 3). Reduced depletion and nighttime capillary water movement into the rootzone then allows greater morning Gs longer into a period of water stress and maintain a degree of photosynthesis, as was apparently the case in this study. Rapid stomatal closure moderating internal water potential in response to water stress to "Wife" is consistent with isohydric behavior on the hydric behavior continuum [13,24].

More notably, quick response of "Wife" to water stress resulted in partial defoliation. Reduced leaf area is a common response in perennial plants, such as trees, to water stress prolonged over the years [25]. Partial defoliation is also a normal phenological response to seasonal drought, such as in *Artemesia tridentata* in the western U.S. steppe region [26]. Defoliation has been reported in fiber hemp in response to water stress [16] and may be viewed in this study as a phenological adaptation to drought. An advantage of partial leaf defoliation is greater illumination of interior leaves, where these remaining leaves would be more efficient in the use of sunlight [27]. Indeed, partial defoliation resulting in fewer but better-illuminated leaves may also explain in part why dry 'Wife' achieved flower yield on par with wet 'Wife' and 'Cherry'.

Conversely, dry-treatment 'Cherry' maintained a full canopy and more open stomata during the dry down, but at the cost of more negative water potential and flower yield, about 20% less than wet-treatment plants. This more aggressive stomatal behavior falls closer to anisohydric behavior on the isohydric-anisohydric continuum [24]. Cannabis germplasm with anisohydric-leaning behavior, and maintaining full canopy in response to

water stress, suggests that water stress can be a major factor in limiting hemp photosynthesis [16] and yield.

Rapid leaf shedding by "Wife" in response to water stress changes the depth of water use substantially. The transpiration weight of dry 'Wife' may have been less than half that of the wet plants, but with half the leaf area defoliated, the normalized water use rate for dry-treatment 'Wife' achieved essentially the same water use rate as wet-treatment plants per unit area (Figure 4F). This would also partially explain the flower yield of dry 'Wife' comparable to wet plants: with half the leaf area, less applied water was enough to keep stomates open and photosynthesis high [26], together with fewer but fully illuminated leaves after defoliation [27]. Reciprocally, "Cherry" water use also fell, but likely due to stomatal closure and at the cost of less flower yield.

The cannabinoid concentrations reported here are largely consistent with the findings of previous research of various hemp cultivars [10,13] where THC concentration decreased with water stress. Indeed, previously published hemp germplasm screening trials have concluded THC (and CBD) to be largely under genetic rather than environmental control [1,10]. There has been research that reported increased THC with water stress for a high-THC cannabis cultivar, but this result could be a trait unique to high THC lines [9,19]. In turn, cultivar differences in flower yield reported here are consistent with previous research that has reported wide genetic variation in yield response to drought treatments, which probably ties back into variation in water use that, again, echo the results of this study [1,2,11].

Understanding hemp water use is important for efficient water management in production situations. Relating non-stressed hemp water use here to ETo would suggest provisional crop coefficients (Kc) of around 1 for 'Cherry', consistent with Kc values found in previous research [11,13], and 1.3 to 1.5 for wet 'Wife'. Further water use studies in field settings would be needed to verify these suggestions and develop best management irrigation practices for larger-scale commercial production.

## 5. Conclusions

There are several takeaways from this study regarding operational practices for hemp production within the context of wide genetic variation. One, while the THC concentration of these two cultivars exceeded the federal threshold of 0.3%, drought stress appears unlikely to worsen (increase) existing THC concentrations. Another is that hemp is a high-water plant, but how high is likely to vary among hemp germplasm, reinforcing previous findings [1,2,11]. Further screening and lysimeter studies are needed in different climates to fine-tune crop coefficients for more precise irrigation scheduling.

A related water use takeaway is deficit irrigation using the distinctive stress-induced leaf shedding by 'Wife'. Deficit irrigation is a management tool to reduce water consumption by imposing mild water stress to partially close stomates and limit transpiration while not limiting photosynthesis [28]. Deficit irrigation would be difficult for the somewhat anisohydric-leaning response of 'Cherry' as it would be a very narrow window of water deficit for partial stomatal closure to balance reduced transpiration with photosynthesis—and not even including possible turgor-related effects on flower yield. The partial defoliation water stress response then would appear to allow gas exchange recovery and presumably photosynthesis. The defoliation response, as seen here for 'Wife', would make an easily detectable deficit irrigation target because of obvious visual cues that could make fine-tuning photosynthesis easier, as well as reducing the extent of vegetation management post-harvest.

However, the genetic unknowns for these takeaways are substantial because cannabis is a very diverse genus [2] and because it is a new crop, standardized genealogy tracking is still rudimentary. The cultivar name 'Cherry', in particular, obscures more than clarifies, given that it appears to be widely crossed with other hemp germplasm [4,22]. More confounding, hemp cultivars with the same name but different commercial sources have been shown to differ in response to photoperiod [18], nutrient availability [8], and rooting propagation [1,2]. Again, further research is needed to link genetic information or identification with a given plant's environmental responses. Further, water stress response

may vary with germplasm based on end-use. In a greenhouse water stress study of a seed-producing hemp cultivar, Gill et al. [29] reported extreme daytime stomatal closure in response to moderate water stress—approximately -1.5 MPa both predawn and midday water potential—suggesting an isohydric response similar to Cherry. However, their reported daytime stomatal conductance values were 20–30% of the values reported here (using the same model porometer from Delta-T Devices). This discrepancy poses the question of hemp cultivars developed for field production of seed, and possibly fiber, having been selected for much more conservative water use on a per-plant basis as opposed to cannabis bred for the production of secondary compounds. While lower Gs in a seed/fiber hemp variety may translate to lower water use per plant than hemp selected for secondary compounds, absolute water use of seed/fiber on a unit area basis may still be quite high because of much greater planting density.

Alternatively, commercial hemp producers can invest the time to develop or adopt practices that identify the needs of the specific hemp cultivars and not assume a broad application of water needs applies to all commercially available cultivars. As documented in this study, the knowledge that such drastic variations in hemp water stress response exist can serve to provide a deeper understanding of the liability a commercial producer may observe when cultivating this unique new specialty crop.

**Author Contributions:** Conceptualization, H.D., R.K., S.A. and B.P.; methodology, H.D. and R.K.; validation, R.K. and B.P.; formal analysis and investigation, H.D., E.B.; resources, S.A.; data curation, H.D. and R.K.; writing—original draft preparation, H.D. and R.K.; writing—review and editing, R.K.; visualization, R.K. and B.P.; supervision, S.A.; project administration, B.P.; funding acquisition, B.P. All authors have read and agreed to the published version of the manuscript.

**Funding:** This research was supported by the Florida Agricultural Experiment Station/National Institute of Food and Agriculture project # FLA-MFC-006143, "Evaluation of production practices on yield and quality of edible and medicinal crops".

**Institutional Review Board Statement:** Not applicable.

**Informed Consent Statement:** Not applicable.

**Data Availability Statement:** Data is available upon request from the corresponding author (R.K.).

**Acknowledgments:** We acknowledge the Florida Agricultural Experiment Station and Institute for Food and Agricultural Sciences for their financial and infrastructure support for this research, and the University of Florida College of Pharmacy for conducting cannabinoid analyses.

**Conflicts of Interest:** The authors declare no conflict of interest.

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
