# Peer review of "Variation in Hydric Response of Two Industrial Hemp Varieties (Cannabis sativa) to Induced Water Stress"

_horticulturae, doi:10.3390/horticulturae9040431_

Round 1

Reviewer 1 Report

1. abstract: 

Line 11-12, Water stress did not affect THC and CBD?  Not significant?

2. abstract: authors need add a research significance of this paper at the last sentence.

3. line 28, Cannabis sativa, should be italic

4. line 39, why authors just refer to “A number of studies have reported responses to N rates”?

5. line 122, Please provide more detailed lighting conditions, such as light intensity

6. 2.3 analysis, this title is too large, maybe THC and CBD content test

7. add a section of data analysis in this paper : 2.4 

8. authors should add more information in the caption of figure, for example, mean, SD, N, error bar etc.

9. In the results of this paper, there are many contents of discussion, and many references are used in the results, and the content of the references are usually used in the discussion

10. The description of analysis of variances are not seen in this article, also nor are analysis of variances and multiple comparisons seen in all pictures.

11. please add a section of “conclusion” in this paper. 

12. There are many problems with the format of references

Author Response

  1. abstract: Line 11-12, Water stress did not affect THC and CBD?  Not significant? Added ‘significant’ to modify ‘affect’
  2. abstract: authors need add a research significance of this paper at the last sentence. Research implications mentioned in abstract lines 21 and 24, and line 24 rewritten to emphasize the need for further germplasm screening and research.
  3. line 28, Cannabis sativa, should be italic. Italicized
  4. line 39, why authors just refer to “A number of studies have reported responses to N rates”? ‘Optimum N fertilization rates is an important environmental input to hemp production with but was not within the research scope here.
  5. line 122, Please provide more detailed lighting conditions, such as light intensity. 110,000 lumens added
  6. 2.3 analysis, this title is too large, maybe THC and CBD content test. Shortened
  7. add a section of data analysis in this paper : Added, but numbering removed.
  8. authors should add more information in the caption of figure, for example, mean, SD, N, error bar etc. More statistical information added to figures where lacking.
  9. In the results of this paper, there are many contents of discussion, and many references are used in the results, and the content of the references are usually used in the discussion. ‘Separate results section parsed out.
  10. The description of analysis of variances are not seen in this article, also nor are analysis of variances and multiple comparisons seen in all pictures. In statistical analysis section more description added for ANOVA and post-hoc test. For the graphs we chose to go with pooled standard deviation to indicate variation among treatments to avoid clutter of the data lines.
  11. please add a section of “conclusion” in this paper. ‘Added
  12. There are many problems with the format of references. (1) (2) Edited citations to comply with Horticulturae guidelines

Author Response

The presented manuscript: "Variation in hydric response of two industrial hemp varieties (Cannabis sativa) to induced water stress" could have scientific and practical relevance for commercial hemp production because the study is focused on the evaluation of the effects of water stress of two hemp varieties on water relations, flower yield, water use and cannabinoid concentrations. However, the manuscript needs extensive revision due to technical, organizational, statistical and scientific issues that should be clarified. The following changes are recommended:

Abstract

Page 1, Line 11, Pg. 2, Line 48: Please, define the abbreviations CBD and THC in the abstract section, and later in the Introduction. Defined

Page 1, Line 21-23: The last sentence of the Abstract is not clear. Please rephrase the following statement: "Because hemp is genetically diverse, and cultivar naming conventions currently lax, the extent to which either conservative ‘Cherry’ or phenologically flexible ‘Wife’ water stress response is found in the larger population of hemp cultivars is unknown until parsed through further germplasm screening and research". Sentence rewritten to emphasize the need for further germplasm screening and research

Introduction

Page 1, Line 28: Please use italic for Cannabis sativa. Italized, line 29

Page 2, Line 52-55: Please rephrase the following sentence: "However, the extent internal water potential and photosynthesis are mediated by stomatal aperture in response to dry air and dry soil along the isohydric/anisohydric continuum [13] is not informed by the extant literature." Rewritten to state that the degree of isohydric/anisohydric behavior has not been reported in hemp

Page 2, Line 63: Please define the abbreviation ETo, since only ET for evapotranspiration is defined. Defined, line 64)

Page 3, Line 47-49: The authors did not adequately respond to the last two topics of this study: 3) What is the effect of stress on flower yield and concentrations of key cannabinoid compounds of interest, particularly THC and CBD; 4) What is the genetic variation in yield, concentration, and water use/relations amongst hemp genetic material? The results for cannabinoid concentrations are not clearly presented and discussed related to the applied stress. In addition, the genetic analysis of the cultivars used in this study was not performed here, in order to relate the analyzed parameters between different genetic materials.  Not sure of point here, since the genetic variation is implicit in the two varieties, and we do note that actual genetic distance between the two cultivars can be inferred from the Phylos Galaxy citation, and the conclusion addresses the need for research to assess the extent that the two water response behaviors are found in other hemp cultivars.  Regarding cannabinoid compounds, there were no differences in concentrations between cultivars and water stress levels, so we are not sure what else is there to say.  What we have done is parsed out a results section, the lack of which was mentioned by other reviewers, that presents the data only, followed by the discussion section that includes citations.

Materials and Methods

Page 4, Line 180-183: The whole part for identification and quantification of cannabinoids in the samples by the UPLC/MS method is missing. Please, briefly describe the chromatographic conditions, such as: type of column, composition of mobile phase, linear or gradient flow rate, injection volume, mass detector conditions, standard used for identification etc. How the quantification of cannabinoids was performed? We understand the expectation of analytical rigor, but this section has been rewritten to clarify that cannabinoid analysis was run in a UF lab following national standards where a complete description of specific details can be found in Berthold et al [22]

Page 4, Line 184-191: Please include separate subdivision in the section Materials and Methods denoted as Statistical analysis and clearly present the data analysis by the statistical methods. How the data are presented, as mean values with standard deviation? How many replicates are used for the analyzed parameters? Done.

Results

Page 4, Line 196: Please describe the abbreviation VPD, since it is mentioned here for the first time in the manuscript. Defined

Page 4, Figure 2: Please denote the corresponded cultivar on the graphs C and D. Please describe in the Figure 2 Caption what is presented on each graph A, B, C and D. This should also be implemented in Figure 3 Caption for graphs A and B.  d

Page 7-8, Table 1: Please, delete the "Total leaf area per plant, specific leaf area and flower yield and yield index", since these parameters are already noted in the Table 1: Rewrote table caption to more completely describe variables reported. Total leaf area per plant may be included in the table instead of leaf area. Done Yield index calculation may be described in the section Materials and methods. What is measurement unit for flower yield, grams per…?  Please define yield index, does it means flower yield index: Explained.

Please, rephrase the following statement under the Table 1: "Values within a column separated by letters are significantly different at p<0.05", since it is not emphasized that different letters denoted significant difference. Rewrote to “…different letters..”). Why the authors did not present the standard deviation along with mean values? Showing both sd and mean separation increases clutter so reduces readability while not adding more information.

Page 9, Figure 4: Please, denote the corresponded cultivar on each graphs (A,B,C,D,E and F). Done.

Page 9, Line 358-376: Why the values for THC and CBD yield are not clearly presented, the authors used "above 0.3%", "slightly greater", "10-11%"? How many replicates for cannabinoid analysis were performed? Our apologies.  Table 2 showing cannabinoid concentrations was inadvertently omitted when manuscript was reformatted to Horticulturae guidelines.  The table has been added.

The sections Results and Discussion should be completely revised. In the section Results, the authors frequently compare the results with the literature data instead of concise and precise description of their experimental results. Several statement and comparison to other studies in the section Results should be included in the section Discussion.  Rewrote to better separate results from discussion with citations.

Reviewer 3 Report

The manuscript presents interesting and useful information, not only for academia but also for farmers. The applied methods are well described, and discussion has a good support from literature. Adding to this, relevant and constructive criticism is made to the present literature regarding how water stress studies and conclusion are taken,  which should be considered a strong point in the manuscript. In my opinion, the present work can be published in the present form after minor revisions. If possible, figures 1 and 2 quality should be improved, and minor English revisions should be taken.

Author Response

The manuscript presents interesting and useful information, not only for academia but also for farmers. The applied methods are well described, and discussion has a good support from literature. Adding to this, relevant and constructive criticism is made to the present literature regarding how water stress studies and conclusion are taken,  which should be considered a strong point in the manuscript. In my opinion, the present work can be published in the present form after minor revisions. If possible, figures 1 and 2 quality should be improved, and minor English revisions should be taken. Thank You

Reviewer 4 Report

The manuscript addresses an interesting topic of practical relevance in the agricultural management of Cannabis sativa. However, I consider that the main limitation is the low number of plants with which the authors extract data and draw conclusions. As the authors describe in materials and methods that they worked with four plants of each variety to apply each of the treatments. Neither did the authors repeat the experimental test to corroborate the results obtained.

Another important shortcoming of the manuscript is that the experimental design is not presented. Statistical analyzes are also not adequately described. Have the ANOVA requirements been tested? With what statistical analysis were the differences between the means of the samples analyzed?

Other comments

Line 11: the authors must indicate what the abbreviations CBD and THC correspond to?

Line 28: Cannabis sativa L. should be written in italics.

Subheadings should not be placed in the introduction in order to make the reading more fluid.

Line 55: Gs units should be mmol m-2s-1

In Fig. 1. temperature scale in °C is not correct.

Line 216: Is the reported value 1.0 or -1.0 MPa?

Fig. 2. On which days were there significant differences in water potential and Gs between treatments? Were there significant differences between the days?

Please be consistent AM/PM or a.m., p.m.

Lines 239-240: In Fig. 2, day 10 is not represented.

Do the authors not show the results of THC and CBD?

Author Response

The manuscript addresses an interesting topic of practical relevance in the agricultural management of Cannabis sativa. However, I consider that the main limitation is the low number of plants with which the authors extract data and draw conclusions. As the authors describe in materials and methods that they worked with four plants of each variety to apply each of the treatments. Neither did the authors repeat the experimental test to corroborate the results obtained. Single-plant reps are common in crops where individual plants are high value, such as fruit trees and here with individual high value high cannabinoid plants, and four reps is common for studies on high value plants, and is enough to run statistical analysis. 

Another important shortcoming of the manuscript is that the experimental design is not presented. Statistical analyzes are also not adequately described. Have the ANOVA requirements been tested? With what statistical analysis were the differences between the means of the samples analyzed? Statistical layout and tests more completely described in separate statistical analysis section in Methods.

Other comments

Line 11: the authors must indicate what the abbreviations CBD and THC correspond to?  Changed

Line 28: Cannabis sativa L. should be written in italics. Italicized

Subheadings should not be placed in the introduction in order to make the reading more fluid. Subheads removed

Line 55: Gs units should be mmol m-2s-1.  Changed

In Fig. 1. temperature scale in °C is not correct. Fixed

Line 216: Is the reported value 1.0 or -1.0 MPa? Rewritten to say that LWP was more negative than -1.0 reported elsewhere.

Fig. 2. On which days were there significant differences in water potential and Gs between treatments? Were there significant differences between the days? Pooled variance shown in graphs allows comparison among treatments.

Please be consistent AM/PM or a.m., p.m. AM/PM changed to a.m/p.m.

Lines 239-240: In Fig. 2, day 10 is not represented. “Wife” Gs and LWP not collected on day 10 because of significant decline on day 8 based on criteria described in Methods.

Do the authors not show the results of THC and CBD? Major error on our part omitting Table 2, cannabinoid yields; table added.

Round 2

Reviewer 1 Report

The author has finished he suggestions, and agrees to accept this paper.

Author Response

No comments, no response

Reviewer 2 Report

The authors did not adequately responded to the following suggestions:

Please, rephrase the following statement under the Table 1: "Values within a column separated by letters are significantly different at p<0.05", since it is not emphasized that different letters denoted significant difference. Rewrote to “…different letters..”). Why the authors did not present the standard deviation along with mean values? Showing both sd and mean separation increases clutter so reduces readability while not adding more information.

Also, there are still many statements and citations in the section Results.

The sections Results and Discussion should be completely revised. In the section Results, the authors frequently compare the results with the literature data instead of concise and precise description of their experimental results. Several statement and comparison to other studies in the section Results should be included in the section Discussion.  

Author Response

Please, rephrase the following statement under the Table 1: "Values within a column separated by letters are significantly different at p<0.05", since it is not emphasized that different letters denoted significant difference. Rewrote to “…different letters..”). Why the authors did not present the standard deviation along with mean values? 

Rewritten

Showing both sd and mean separation increases clutter so reduces readability while not adding more information.

SD added to both Table 1 and 2.

Also, there are still many statements and citations in the section Results. The sections Results and Discussion should be completely revised. In the section Results, the authors frequently compare the results with the literature data instead of concise and precise description of their experimental results. Several statement and comparison to other studies in the section Results should be included in the section Discussion.

All discussion points with citations have been removed from Results and into Discussion

Reviewer 4 Report

The manuscript improved compared to its previous version.

The units of Gs are mmol m-2s-1, correct at line 223 and in Fig. 1

The authors should improve the Statistical Analysis section. Detail the statistical analyzes that were carried out to determine if there were  differences in water potential, stomatal conductance, etc., between sampling times and cannabis varieties.

In results, the authors must attach tables with the results of the statistical parameters for the variables water potential, stomatal conductance, volumetric content of soil water, water use, etc. for the effects of sampling time, species, and interaction.

Author Response

The units of Gs are mmol m-2s-1, correct at line 223 and in Fig. 1.  Corrected

The authors should improve the Statistical Analysis section. Detail the statistical analyzes that were carried out to determine if there were  differences in water potential, stomatal conductance, etc., between sampling times and cannabis varieties. Further description of t-test and post-doc analysis of Figures 1-3 added. 

In results, the authors must attach tables with the results of the statistical parameters for the variables water potential, stomatal conductance, volumetric content of soil water, water use, etc. for the effects of sampling time, species, and interaction. Have added statistics to Figures 1-3, but please note that statistics for a physiological input (Gs, LWP, or water use) has less meaning than appropriate statistical tests for a response such as biomass yield or cannabinoid concentration.  Physiological input responses such as Gs, LWP, or water use have to be interpreted as a time series (hour, day) pattern, and not where one day or hour is significantly different among treatments.     Adding tables with more statistics would be redundant with stats presented in the figures 1-3, so not a good use of space.  We have not added further statistics to water use data because anything added to Figure 4 above the pooled standard deviation would add clutter and complexity, and so reduce readability and interpretation of the time series as an integrated patterns.  Thus we argue that the pooled standard deviation shown in Figure 4 is sufficient.